

# Scalable imaginary time evolution with neural network quantum states

**Eimantas Ledinauskas[1,2]⋆ and Egidijus Anisimovas[1]**

**1** Institute of Theoretical Physics and Astronomy, Vilnius University,
Saulėtekio al. 3, LT-10257, Vilnius, Lithuania
**2** Baltic Institute of Advanced Technology, Pilies St. 16-8, LT-01403, Vilnius, Lithuania

⋆ eimantas.ledinauskas@ff.stud.vu.lt

## Abstract

The representation of a quantum wave function as a neural network quantum state (NQS) provides a powerful variational ansatz for finding the ground states of many-body quantum systems. Nevertheless, due to the complex variational landscape, traditional methods often employ the computation of quantum geometric tensor, consequently complicating optimization techniques. Contributing to efforts aiming to formulate alternative methods, we introduce an approach that bypasses the computation of the metric tensor and instead relies exclusively on first-order gradient descent with Euclidean metric. This allows for the application of larger neural networks and the use of more standard optimization methods from other machine learning domains. Our approach leverages the principle of imaginary time evolution by constructing a target wave function derived from the Schrödinger equation, and then training the neural network to approximate this target. We make this method adaptive and stable by determining the optimal time step and keeping the target fixed until the energy of the NQS decreases. We demonstrate the benefits of our scheme via numerical experiments with 2D $J_1$–$J_2$ Heisenberg model, which showcase enhanced stability and energy accuracy in comparison to direct energy loss minimization. Importantly, our approach displays competitiveness with the well-established density matrix renormalization group method and NQS optimization with stochastic reconfiguration.

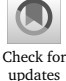

# 1 Introduction

The remarkable accomplishments in applying machine learning techniques to a wide range of practical [1–6] and curiosity-driven [7–9] tasks have prompted the adoption of innovative concepts and approaches in the field of physical sciences; see e.g. Refs. [10–15] for recent physics-motivated reviews and tutorials. In particular, in the numerical study of quantum many-body systems [16] one promising line of development rests on the observation that, given a suitable many-body basis, a pure quantum-mechanical state can be represented by a multivariate function that assigns a complex number to a given basis element. For a typical quantum state of interest, such a wave function is an overwhelmingly complex entity, as the amount of information needed to fully encode a state in this manner grows exponentially with the system size. On the other hand, it is fitting to note that many classes of neural network architectures are recognized [17–21] as efficient function approximators: an arbitrary well-behaved function can be represented with a desired accuracy given a sufficient number of parameters, i.e. a sufficient neural network width or depth. Note, however, that such *universal approximation theorems* [20] only establish that a representation of a given function is possible, without providing an explicit scheme to construct such a representation.

The idea to represent a quantum wave function as a neural network quantum state (NQS) [11, 15, 22–28], can be viewed as a very powerful variational ansatz that aligns well with the well-established variational Monte-Carlo (VMC) [29, 30] techniques to search for the ground state. In the general application framework, the approach typically includes the following elements: (I) The variational wavefunction $\psi_\theta$ is parameterized in terms of the network weights $\theta$, and the expectation value of the studied Hamiltonian $H$ (the variational energy) $E(\theta) = \langle \psi_\theta | H | \psi_\theta \rangle$ is identified as the objective function to be minimized, i.e. the *loss* function. (II) Since a direct computation of $E(\theta)$ is not feasible due to prohibitively large configuration space, $E(\theta)$ is estimated as a stochastic average from a set of sampled configurations. Note that sampling can be performed either relying on a Markov chain of configurations [22, 31] or direct sampling from a neural network endowed with the autoregressive property [25, 26, 32].

(III) Since the variational landscapes of NQS states are rugged and difficult to navigate in the search of the ground state (see, e.g., the study reported in Ref. [31]), the stochastic reconfiguration (SR) method [33, 34] is often used instead of some variant of direct gradient-based descent [35]. According to the geometric interpretation [36, 37], the benefits provided by the SR method stem from the correct determination of the non-Euclidean manifold structure of the space of quantum states. Thus, the minimization direction suggested by the raw gradient, $-\nabla_\theta E(\theta)$, is less meaningful than the 'natural' Riemannian gradient [37, 38] corrected by the inverse of the local metric tensor $G(\theta)$. The educated update of parameters is made along the direction of $-G^{-1}(\theta)\nabla_\theta E(\theta)$. However, while using the metric tensor may improve the convergence, it comes with a substantial computational cost since the metric tensor is of the order $N_\theta \times N_\theta$, where $N_\theta$ represents the number of NQS variational parameters. Consequently, determining and calculating the (pseudo)inverse of the metric tensor becomes computationally expensive [10] and results in poor scaling with the system size [39]. This issue imposes significant limitations on the practical size of neural networks that can be utilized. The scaling of SR can be enhanced by employing iterative solvers that avoid forming or inverting the tensor (e.g., see the implementation of SR in NetKet [40, 41]). However, this still adds a significant computational cost and also increases the numerical instability, which necessitates adding a small diagonal shift to stabilize the method, which, in turn, reduces the precision of the estimated metric.

An alternative approach emerged recently, in which energy is not minimized directly. Rather, a target wave function, more proximate to the ground state, is crafted by employing the approximate imaginary time evolution and the issue is reformulated into a conventional regression problem [42–44]. In the present paper, we focus on this alternative methodology, refining and integrating modifications to shape it into a novel scheme that enhances its performance and stability. Firstly, we employ the more natural overlap loss function instead of the mean square error loss function, as introduced in works concerned with the real-time evolution [45–47]. Secondly, we derive the expression for the adaptive optimal imaginary time step. Thirdly, instead of updating the target wave function after each step, we fix the target for multiple steps to stabilize the learning process and establish criteria for determining when the target should be updated. We also explore the relationship between our scheme and direct energy minimization, demonstrating that they become equivalent when the target wave function is updated after every time step. This may elucidate why utilizing energy as a loss function does not always work well and requires more sophisticated methods like SR. Our simulations indicate that, with our proposed optimization scheme, the NQS is able to avoid being trapped in the numerous saddle points of the optimization landscape, eliminating the need for metric tensor computations and operating effectively with solely first-order gradient descent. We validate the efficacy of this scheme by applying it to the two-dimensional $J_1$–$J_2$ Heisenberg model [48, 49], demonstrating that the proposed approach is competitive with direct energy loss minimization using first-order gradients or SR, as well as with density-matrix renormalization (DMRG) [50, 51].

Our paper is organized as follows. Section 2 provides an overview of the proposed scheme for ground-state search and describes the neural network optimization procedure. While the process assumes that the state is represented by a neural network, it is versatile and can be used with diverse network architectures. Thus, in Section 3 we describe the specific architecture used in our work. It is based on a multilayer perceptron (MLP) [52] coupled with a preprocessing step inspired by vision transformers [3, 6]. The physical model used for testing, i.e. the two-dimensional spin-$\frac{1}{2}$ $J_1$–$J_2$ model, and the results of numerical experiments are discussed in Section 4. Finally, we conclude with a brief summarizing Section 5. A number of technical derivations are included in Appendices.

## 2 Proposed method for ground state search with NQS

### 2.1 Problem definition and notation

The focus of our work is the ground-state search for many-particle quantum systems defined on a lattice. More specifically, we study two-dimensional (2D) square lattices, however, the described methods can be straightforwardly generalized to other lattice geometries and dimensionalities. In such systems, the space of quantum states can be modeled as a Cartesian product of identical single-site state spaces $\mathcal{H} = \bigotimes_{j=1}^{N} \mathcal{H}_1$, where $N$ is the number of lattice sites and $\mathcal{H}_1$ is the state space characterizing a single site.

We choose to work in the product basis $|\boldsymbol{s}\rangle = \bigotimes_{j=1}^{N} |s_j\rangle$, where $|s_j\rangle$ corresponds to a basis vector of $\mathcal{H}_1$ on the $j$-th site. These basis states can be indexed with $N$-tuples $\boldsymbol{s} = (s_1, ..., s_N)$, thus we can introduce the notation $|\boldsymbol{s}\rangle = |s_1, ..., s_N\rangle$. For example, in the case of spin-1/2 particles occupying lattice sites we can choose $|s_j\rangle \in \{|\uparrow\rangle, |\downarrow\rangle\}$, where $|\uparrow\rangle$ and $|\downarrow\rangle$ are the eigenstates of the spin operator $\hat{S}_z$. In this basis, the wave function is represented as a complex-valued vector

$$|\psi\rangle = \sum_{s_1, ..., s_N} \psi_{s_1, ..., s_{N_s}} |s_1, ..., s_N\rangle. \tag{1}$$

The dimension of the vector space spanned by the basis vectors $|s_1, ..., s_N\rangle$ grows exponentially with increasing system size and rather quickly it becomes impractical to numerically represent and manipulate the wave function directly as an array. To model such directly intractable systems, Ref. [22] proposed using a neural network which maps basis vectors $|s_1, ..., s_N\rangle$ to the corresponding wave function amplitudes $\psi_{s_1, ..., s_N}$; this representation has become known as the NQS. The overarching motivation stems from the observation that in spite of the exponential scaling of the number of basis elements, typical ground states of physically realizable Hamiltonians have a simplified internal structure [51] that should allow for representations in terms of a relatively small number of parameters. On the other hand, neural networks excel precisely at the task of finding efficient representations of intricate data structures.

### 2.2 Imaginary time evolution with NQS

Imaginary time evolution is a well-known method for extracting the ground state from an arbitrary initial ansatz that is not strictly orthogonal to the sought ground state. In the energy-eigenstate basis the time-dependent Schrödinger equation is solved by:

$$|\psi\rangle = \sum_j \Psi_j(t)|e_j\rangle, \tag{2}$$

with

$$\Psi_j(t) = \Psi_{j,0} e^{-iE_j t}, \tag{3}$$

where $E_j$, $|e_j\rangle$ are the $j$-th eigenvalue and eigenvector pair of the Hamiltonian $\hat{H}$ and the amplitudes $\Psi_{j,0}$ encode an arbitrary initial condition. If we now substitute in the imaginary time $\tau = it$, then with increasing $\tau$ the solution is expressed as an exponentially decaying (rather than the usual oscillating) superposition of energy eigenstates. The contributions of all the excited states decay exponentially relative to the ground state $|e_0\rangle$:

$$\frac{|\Psi_j|}{|\Psi_0|} \propto e^{-(E_j - E_0)\tau}. \tag{4}$$

Thus, as $\tau \to \infty$, any initial state with nonzero overlap with $|e_0\rangle$ converges toward the ground state. It is important to note that with imaginary time the evolution becomes non-unitary and the norm of the wave function increases or decreases exponentially with time.

In Ref. [28], the authors demonstrate that real time evolution of a quantum system can be modeled with NQS by minimizing the error between the variational wave function and the target wave function which is obtained with discrete ODE solver:

$$\left\| |\psi_{m+1}\rangle - \Phi^{\Delta t} |\psi_m\rangle \right\|, \tag{5}$$

where $\Phi^{\Delta t}$ is the discrete ODE flow operator. Minimizing this error draws the variational wave function closer to the target, computed approximately from the Schrödinger equation, thereby simulating the temporal evolution of the NQS. A similar approach has also been applied to simulate imaginary time evolution [42–44].

In this work, we use the Euler method to compute the target wave function:

$$|\psi_T\rangle = |\psi\rangle - \Delta\tau \hat{H} |\psi\rangle, \tag{6}$$

that is, the wave function that is reached from the current wave function by a single linearized imaginary time evolution step $\Delta\tau$. We note that for Hamiltonians consisting of only local terms, the matrix $H_{\mathbf{ss}'} = \langle \mathbf{s} | \hat{H} | \mathbf{s}' \rangle$ is sparse, i.e., given some $\mathbf{s}'$, only a small number of elements $H_{\mathbf{ss}'}$ are non-zero. This enables the efficient computation of the target wave function

$$\psi_T(\mathbf{s}) = \psi(\mathbf{s}) - \Delta\tau \sum_{\mathbf{s}'} H_{\mathbf{ss}'} \psi(\mathbf{s}'), \tag{7}$$

from the current state of the neural network. We perform a single time step to obtain a target and then train the variational NQS. Performing a second time step is not practical for large systems, as it would necessitate an additional application of the Hamiltonian to the resulting target wave function. This would change the copmutational complexity scaling from linear to quadratic with respect to system size (as similarly described in Eq. 18).

As NQS cannot perfectly fit the target function, the resulting evolution is inherently noisy. Nevertheless, Ref. [43] has established a convergence guarantee for this type of evolution.

## 2.3 Loss function

Multiple different loss functions can be used to maximize the consistency between the current wave function, $|\psi\rangle$, and the target wave function, $|\psi_T\rangle$. We experimented with several variants and found that the following loss function based on overlap works well in practice:

$$L = -\log\left[ \frac{|\langle \psi | \psi_T \rangle|^2}{\langle \psi | \psi \rangle \langle \psi_T | \psi_T \rangle} \right], \tag{8}$$

where $|\psi\rangle$ denotes the NQS to be optimized and $|\psi_T\rangle$ denotes the target constructed by Eq. (6). This kind of overlap loss function has been employed in previous studies within the context of real-time evolution (e.g. [45–47]). In the subsequent text, we will refer to this loss together with the target $|\psi_T\rangle$ as the *ITE loss*. The gradient of ITE loss with respect to the variational NQS parameters $\theta$ can be estimated by using the following expression (derivation is presented in the appendix, Sec. B):

$$\frac{\partial L}{\partial \theta} = 2\Re \left\{ \left\langle \frac{\partial}{\partial \theta} \log \psi^*(\mathbf{s}) \right\rangle_{\mathbf{s}} - \frac{1}{\left\langle \frac{\psi_T(\mathbf{s})}{\psi(\mathbf{s})} \right\rangle_{\mathbf{s}}} \left\langle \frac{\psi_T(\mathbf{s})}{\psi(\mathbf{s})} \frac{\partial}{\partial \theta} \log \psi^*(\mathbf{s}) \right\rangle_{\mathbf{s}} \right\}. \tag{9}$$

The averages appearing in this equation can be estimated by sampling states, $\mathbf{s}$, according to the probability distribution, $|\psi(\mathbf{s})|^2$, and then utilizing $\langle f(\mathbf{s}) \rangle_{\mathbf{s}} \approx \frac{1}{N} \sum_{\mathbf{s}_j} f(\mathbf{s}_j)$, where $\mathbf{s}_j$ represents individual states from a finite sample. During optimization Eq. (9) can be used directly

without actually computing the loss. The gradients of NQS $\frac{\partial}{\partial \theta} \log \psi^*(s)$ can be calculated by utilizing the automatic differentiation provided by deep learning libraries.

The intuition behind the approach can be explained as follows: The ITE loss pushes for an alignment between the current wave function $|\psi\rangle$ and the target $|\psi_T\rangle$. On the other hand, the two states are related by an imaginary time step, hence, in the target $|\psi_T\rangle$ all components spanned by low-energy eigenstates are exponentially amplified, and all components spanned by high-energy eigenstates are exponentially suppressed. Now, since the target is kept fixed for a number of optimization steps, the update of the weights of the neural network encoding the current state $|\psi\rangle$ will effectively drive towards the minimization of the state energy, i.e., towards the ground state.

## 2.4 Adaptive imaginary time step

In the case of the Euler's method, Eq. (6) can be used to obtain the following relation between the average energy of the target wave function and imaginary time step, $\Delta\tau$:

$$\langle E_T\rangle = \frac{\langle\psi_T|\hat{H}|\psi_T\rangle}{\langle\psi_T|\psi_T\rangle} = \frac{\langle E\rangle - 2\Delta\tau\langle E^2\rangle + \Delta\tau^2\langle E^3\rangle}{1 - 2\Delta\tau\langle E\rangle + \Delta\tau^2\langle E^2\rangle}, \tag{10}$$

where $\langle E^n\rangle = \langle\psi|\hat{H}^n|\psi\rangle/\langle\psi|\psi\rangle$. Setting $\partial_{\Delta\tau}\langle E_T\rangle = 0$ one obtains a quadratic equation which is solved by:

$$\Delta\tau = \frac{B \pm \sqrt{B^2 + 4A\sigma^2}}{2A}, \tag{11}$$

where $A = \langle E^2\rangle^2 - \langle E\rangle\langle E^3\rangle$, $B = \langle E\rangle\langle E^2\rangle - \langle E^3\rangle$, and $\sigma^2 = \langle E^2\rangle - \langle E\rangle^2$. We use Eq. (11) to find the optimal time step which minimizes the energy of the target wave function. This significantly accelerates the convergence to the ground state. The energy averages required in Eq. (10), similar to the averages appearing in Eq. (9), can be estimated by sampling basis states $s$ according to the probability distribution, $|\psi(s)|^2$, and using the following equations:

$$\langle E\rangle = \langle H_{loc}(s)\rangle_s, \tag{12}$$

$$\langle E^2\rangle = \left\langle |H_{loc}(s)|^2\right\rangle_s, \tag{13}$$

$$\langle E^3\rangle = \left\langle H_{loc}^*(s)\left(H^2\right)_{loc}(s)\right\rangle_s, \tag{14}$$

where $H_{loc}$ is defined by:

$$H_{loc}(s) = \sum_{s'}\frac{\psi(s')}{\psi(s)}\langle s|\hat{H}|s'\rangle, \tag{15}$$

and $\left(H^2\right)_{loc}$ is defined analogously but using $\hat{H}^2$ instead of $\hat{H}$. These expressions are widely known but for the sake of completeness, the derivations are presented in Appendix A. The computation of $H_{loc}$ is efficient because in systems with local interactions (e.g. between nearest neighbors), the Hamiltonians are represented by highly sparse matrices, and only a small fraction of the terms in the sum need to be calculated.

In the case of local interactions the computational complexity of $H_{loc}$ scales linearly with system size and for $\left(H^2\right)_{loc}$ it scales quadratically because it requires computing nested sums of matrix elements:

$$H_{loc}^2(s) = \sum_{s'}\frac{\psi(s')}{\psi(s)}\sum_{s''}H_{ss''}H_{s''s'} \tag{16}$$

$$= \frac{1}{\psi(s)}\sum_{s''}H_{ss''}\sum_{s'}\psi(s')H_{s''s'}, \tag{17}$$

where $H_{ss'} = \langle s|\hat{H}|s'\rangle$. This means that the computation of $\langle E^3\rangle$ quickly becomes impractical with increasing system size. However, by experimenting we found that the following expression gives sufficient accuracy for the optimal time step calculation even though it is not strictly correct:

$$\langle E^3\rangle \approx \left\langle H_{loc}(s)|H_{loc}(s)|^2\right\rangle_s . \tag{18}$$

Consequently, with this approximation, the computational complexity of all required energy moments for our method scales linearly with the system size.

## 2.5 Relation between ITE loss and E loss

In many standard applications of NQS for the ground state search it is performed by directly utilizing the energy as the loss function. In that case, the gradient of the loss with respect to variational parameters is given by:

$$\frac{\partial L_E}{\partial \theta} = 2\Re\left\{\left\langle H_{loc}(s)\frac{\partial}{\partial\theta}\log\psi^*(s)\right\rangle_s - \langle H_{loc}(s)\rangle_s\left\langle\frac{\partial}{\partial\theta}\log\psi^*(s)\right\rangle_s\right\} . \tag{19}$$

In this work, we refer to $L_E$ as *E loss*. By plugging the expression for $|\psi_T\rangle$ from Eq. (7) into Eq. (8) it can be shown that ITE loss with Euler step target becomes proportional to E loss (for derivation see Appendix C):

$$\frac{\partial L}{\partial \theta} = \frac{\Delta\tau}{1-\Delta\tau\langle E\rangle}\frac{\partial L_E}{\partial \theta} . \tag{20}$$

So it would appear that ITE loss does not offer any advantages and it should perform more or less the same as E loss. However, it is only so in the case when the target wave function $|\psi_T\rangle$ is updated after every optimizer step. If instead it is held fixed for more than one step, Eq. (20) does not apply.

The relationship presented in Eq. (20) suggests an explanation for the poor performance of first-order gradient descent methods within the context of NQS ground state search. Using E loss is equivalent to fitting a constantly shifting target described by Eq. (7). This perpetually changing target can destabilize stochastic gradient descent. A similar problem also arises within the context of deep reinforcement learning. For example, it is well established that using fixed targets significantly improves the performance of deep Q learning methods [53]. Similarly, in this work, we also demonstrate in Sec. 4.3 that employing ITE loss with a fixed target can enhance stability and yield lower energy errors compared to E loss.

## 2.6 NQS training procedure

As described in Sec. 2.5, computing $|\psi_T\rangle$ with the latest NQS parameters every optimizer step would make ITE loss equivalent to E loss up to a learning-rate multiplier. There are two simple methods to slow down the shift of the target: 1) using a duplicate neural network with weights given by the moving average of the neural network that is optimized; 2) using a duplicate neural network with fixed weights that are repeatedly updated after a certain number of optimizer steps. Both of these methods are widely used in reinforcement learning literature within the context of action value function learning [54]. We chose to use the second method because in that case it is straightforward to utilize the optimal time step described in Sec. 2.4 and also because it is a more natural fit for evolution with discrete time steps.

We explored various approaches to control the update frequency of the fixed parameters and found that the following approach provides good consistency and efficiency. The target is fixed until the mean energy of the optimized NQS becomes smaller than $\langle E\rangle - \sigma_E$ where $\langle E\rangle$ is the energy of the fixed NQS and $\sigma_E$ is the standard error of the mean-energy estimate.

In this paragraph, we provide a concise summary of the training procedure that simulates the imaginary time evolution. Training is split into multiple epochs where each epoch corresponds to a single discrete time step $\Delta\tau$. At the start of each epoch the energy moments $\langle E \rangle$, $\langle E^2 \rangle$, $\langle E^3 \rangle$ are estimated by sampling $N_E$ states according to the NQS with the latest parameters and utilizing Eqs. (12), (13), and (18). These estimates are then used to calculate the adaptive time step $\Delta\tau$ that minimizes the expected energy of $|\psi_T\rangle\rangle$ given by Eq. (10). The duplicate NQS that is used to compute the target wave function, $|\psi_T\rangle$, is updated only at the start of the epoch and is kept fixed until the next epoch. Conceptually this can be understood as fixing $|\psi_T\rangle$ until the next epoch. During the epoch, the loss function, given by Eq. (8), is gradually minimized with stochastic gradient optimizer (ADAM [55] in our case) so the optimized NQS becomes increasingly similar to $|\psi_T\rangle$ and the energy decreases. The epoch is terminated once the energy becomes smaller than the threshold value described in the previous paragraph. Accurately estimating the energy with a large number of basis state samples can be computationally expensive. To address this, we reuse the samples used in gradient estimation and then apply a running average to the resulting energies, reducing the noise. Please note that we employ this approximate scheme solely for energy estimation during the epoch to compare with the threshold. The threshold itself is computed more accurately with a large number of samples $N_E$, as described in the beginning of this paragraph. A simplified pseudocode of the training algorithm is provided in Alg. 1.

---

**Algorithm 1** NQS training with ITE loss

NQS ← RANDOMINITIALIZATION()
**for** $N_{\text{epochs}}$ **do**
    $\langle E \rangle, \langle E^2 \rangle, \langle E^3 \rangle, \sigma_E$ ← ENERGYSTATISTICS($NQS$)
    $\Delta\tau$ ← OPTIMALTIMESTEP($\langle E \rangle, \langle E^2 \rangle, \langle E^3 \rangle$)
    $E_{\text{thr}}$ ← $\langle E \rangle - \sigma_E$
    $NQS_{\text{fixed}}$ ← COPY(NQS)
    $s_{\text{mcmc}}$ ← WARMUPMCMC(NQS)         ▷ initial batch of basis states for MCMC sampling
    **while** $\langle E \rangle > E_{\text{thr}}$ **do**
        $s_{\text{batch}}, s_{\text{mcmc}}$ ← SAMPLEMCMC(NQS, $s_{\text{mcmc}}$)
        $\psi_{\text{target}}$ ← COMPUTETARGET($NQS_{\text{fixed}}$, $s_{\text{batch}}$, $\Delta\tau$)
        grad ← LOSSGRADIENT(NQS, $\psi_{\text{target}}$, $s_{\text{batch}}$)
        NQS ← UPDATEPARAMETERS(NQS, grad)
        $\langle E \rangle$ ← MOVINGAVERAGEENERGY(NQS, $s_{\text{batch}}$)
    **end while**
**end for**

---

The optimization scheme we presented has similarities with the one described in Ref. [42]. However, we employ a different loss function, utilize different criteria for epoch termination, and incorporate an adaptive time step, which significantly enhances the convergence rate.

## 3 Neural network quantum state

### 3.1 Neural network architecture

The existing literature on NQS encompasses a wide range of neural network architectures. For example, Ref. [56] used multilayer perceptrons (MLPs), Ref. [57] used convolutional neural networks (CNNs), Ref. [25] used recurrent neural networks (RNNs), and Ref. [27] used transformers. In this work, we chose to employ MLPs since our preliminary experiments indicated that they offer the best balance between the required computational resources, simplicity, and

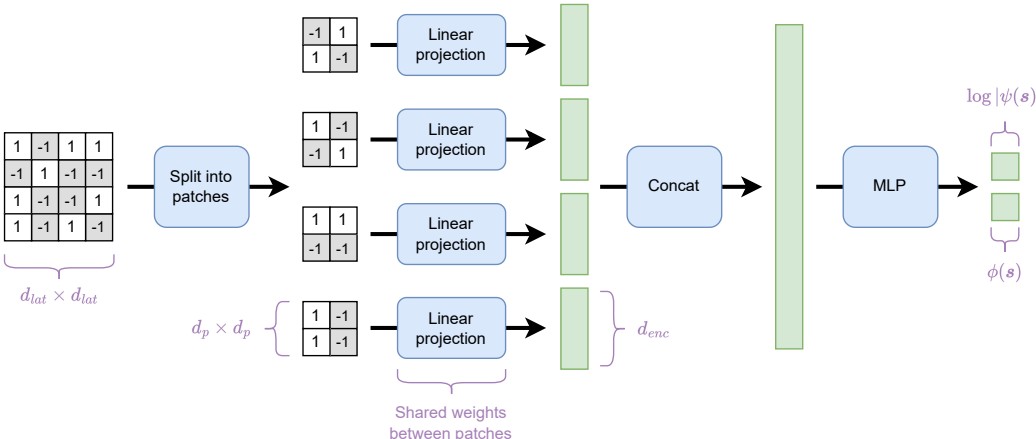

Figure 1: Scheme of the NQS architecture used in this work.

performance. We do not claim that MLPs are the best architecture for NQS and acknowledge that a more detailed analysis is required. However, in this work, our focus is on introducing a novel NQS training method rather than achieving the best possible performance in modeling a specific system. The resulting architecture imposes minimal assumptions on the physical system, enabling its application to cases beyond the scope of this work. This includes diverse lattice patterns, different symmetries, and varying numbers of dimensions.

In the case of 2D spin lattice models the most straightforward way to encode the input basis states would be to apply the following map on lattice site states: $|\uparrow\rangle \rightarrow -1$ and $|\downarrow\rangle \rightarrow 1$. The resulting matrix can be flattened and used as input for the first layer of MLP. However, we discovered that a better performance can be obtained by utilizing the encoding based on 2D patches that is used with vision transformers [3]. The matrix described above is divided into patches of size $d_p \times d_p$, which are subsequently mapped to vectors of dimension $d_{enc}$ using a linear transformation with trainable weights. The resulting vectors are concatenated and then used as input for the first layer of MLP. This kind of encoding has been demonstrated to be beneficial in image processing, not only with transformers but also with other architectures (e.g. [6]).

There are two general approaches for defining the output of the neural network within the context of NQS. The first approach is to simply output a complex number that corresponds to the amplitude of the non-normalized wave function (e.g. [22, 31, 58]). The second approach involves factorizing the wave function into conditional factors, akin to the chain rule of probabilities, and autoregressively generating these conditional wave functions as output (e.g. [25, 26, 32]). In this work, we opted to utilize the first approach. Nevertheless, our method could be applied using the second approach as well.

For the sake of simplicity, we maintain a constant number of neurons across all hidden layers of the MLP. The final layer of the model produces two output values. The first value represents the log-amplitude of the wave function, while the second value represents the phase. Because only the fractions of different wave function elements have meaning, the output amplitudes could grow indefinitely during training. To prevent that we follow Ref. [31] and utilize the following nonlinearity for the log-amplitude value: $f(x) = a \tanh(x/a)$. We set $a = 20$ which allows to express amplitudes within the range of 17 orders of magnitude. For the phase value, we do not apply any nonlinearities and allow values outside the $(0, 2\pi)$ range. All of the variational parameters of the neural network are real. The neural network's architecture is illustrated in Fig. 1.

## 3.2 Sampling of basis states

As described in Sec. 2.3 and 2.4, the estimation of the gradient and energy statistics requires sampling the basis states according to the probability distribution associated with the wave function, $p(\mathbf{s}) = |\psi(\mathbf{s})|^2$. Due to the high dimensionality of the problem and the fact that our NQS outputs unnormalized amplitudes, direct sampling is not feasible. Instead, we employ Markov chain Monte Carlo (MCMC) sampling, specifically the Metropolis-Hastings algorithm [59, 60], which has become a standard practice in the field of NQS.

Here we provide a brief overview of the algorithm specifically for a spin-lattice model with fixed magnetization. Starting basis state, $\mathbf{s}_0$, is sampled according to the uniform distribution. At the $m$-th step a new basis state candidate, $\mathbf{s}'_m$, is generated by randomly choosing a pair of neighbor sites in $\mathbf{s}_{m-1}$ and interchanging their spin states. This candidate is accepted by setting $\mathbf{s}_m = \mathbf{s}'_m$ if $u \leq |\psi(\mathbf{s}')|^2 / |\psi(\mathbf{s})|^2$ where $u \in [0, 1]$ is a random uniform number. If a candidate is not accepted, then the basis state is kept unchanged: $\mathbf{s}_m = \mathbf{s}_{m-1}$. The algorithm proceeds to randomly walk in the sample space and generates samples that eventually start to follow the probability distribution $|\psi(\mathbf{s})|^2$. The initial samples may follow a different distribution. To address this, we incorporate a warm-up period lasting $N_{\text{warmup}} = 10d_{\text{lat}}^2$ steps, during which the generated samples are discarded. To reduce sample correlation, we selectively retain samples at a regular interval of $N_{\text{skip}} = 4$.

To enhance the efficiency of MCMC sampling, we parallelize the algorithm by initializing multiple random basis states and subsequently conducting independent MCMC random walks for each of them. We also perform the warm-up period only at the beginning of the epoch and then reuse the MCMC walker values from the last step as initial values for the current step. These two optimizations increase the training speed by orders of magnitude. In this work, we set the number of MCMC walkers equal to the batch size, $N_{\text{b}}$. Consequently, we only need to evaluate the NQS $N_{\text{skip}}$ times to generate a batch of basis states sufficient for a single optimizer step.

## 3.3 Implementation details

The code was written in Python [61]. Numerically demanding parts were implemented by using JAX [62] package. Neural network computation and training parts were implemented by using FLAX [63] and OPTAX [64] packages. In this work, JAX was useful not only because of its automatic differentiation but also because of its just-in-time compilation and automatic vectorization (*vmap*). With these capabilities, we could easily parallelize and optimize local energy computation and state sampling functions which led to significant computation time improvements.

For SR computations we used the implementation in NETKET [40, 41]. For exact diagonalization (ED) and DMRG computations based on QUSPIN [65, 66] and TENPY [67], we utilized the AMD Ryzen Threadripper 2990WX CPU. NQS training was conducted using the Nvidia RTX 4090 GPU.

# 4 Numerical experiments

## 4.1 $J_1$ - $J_2$ Heisenberg model on a 2D square lattice

Let us now benchmark the performance of the proposed approach by treating a specific numerical example. For this purpose, we choose the two-dimensional $J_1$–$J_2$ Heisenberg model,

defined by the Hamiltonian

$$H = J_1 \sum_{\langle ij \rangle} \vec{S}_i \cdot \vec{S}_j + J_2 \sum_{\langle\langle ij \rangle\rangle} \vec{S}_i \cdot \vec{S}_j \,, \tag{21}$$

with $\vec{S}_j$ denoting the spin-$\frac{1}{2}$ operators defined on the sites (indexed by $j$) of 2D $d_{\text{lat}} \times d_{\text{lat}}$ square lattice with periodic boundary conditions. The model features competing antiferromagnetic interactions: $J_1 > 0$ act between the nearest-neighbor spin pairs ($\langle ij \rangle$), and $J_2 \geqslant 0$ couple the next-nearest neighboring pairs situated on the opposite ends of diagonals of the square plaquettes. In the absence of the second term, these nearest-neighbor interactions stabilize the Néel order with two opposite-magnetization sublattices intertwined in a checkerboard pattern. In the opposite limit of dominant long-range interactions $J_2 \gg J_1$, the model features a striped antiferromagnetic phase. In the vicinity of the classical boundary $J_2/J_1 = 0.5$ the model is frustrated and the exact phase diagram is subject to an ongoing debate, see e.g. Refs. [48, 49, 68] and references therein.

This particular model exhibits several symmetries: 1) translation along the x-axis and y-axis; 2) rotation by multiples of 90 degrees; 3) reflection about the x-axis, y-axis, and diagonal; 4) spin inversion; 5) SU(2) spin rotation. These symmetries can be leveraged to effectively reduce the dimensionality of the problem, as was done in many previous studies (e.g. [31, 58] and references therein). However, in this work we do not enforce the NQS to be invariant with respect to any of the aforementioned transformations. By adopting this approach, we aim to demonstrate that when utilizing large neural networks, there is no need for specialized architectures tailored to specific systems.

## 4.2 Hyperparameters

The values of various hyperparameters, defined in the text, are listed in Table 1. Unless explicitly stated otherwise, these values were consistently applied across all our numerical experiments.

In all our numerical experiments, we optimize NQS using ADAM optimizer with first order gradient descent. The learning rate is adjusted using an exponential decay schedule, with the initial learning rate $\alpha_0$ and the final learning rate $\alpha_f$.

To identify suitable values for the neural network width (number of neurons per hidden layer) and depth (number of hidden layers), we conducted a small grid search. We explored three width values (128, 256, 512) and varied the depth from 1 to 5. As a performance metric, we utilized the achieved energy error (relative to ED results) for a $6 \times 6$ lattice with $J_2/J_1 = 0.5$. The results are illustrated in Fig. 2. It is evident that, in general, accuracy tends to improve with an increase in the number of parameters in the network. This is expected, as larger neural networks possess the capacity to represent a broader subspace of all possible wave functions. In fact, the accuracy does not appear to plateau as the network width increases. It is likely that further improvement in accuracy can be achieved by increasing the network width even more. However, the depth does exhibit an optimal value, as the accuracy tends to decline when the depth exceeds 4. This is also expected since training deeper neural networks necessitates additional techniques such as residual connections [69] or specific initialization [70]. Based on these results, we employ a depth of 4 and a width of 512 in the subsequent numerical experiments, as this configuration demonstrated the best performance. This particular network has 893994 variational parameters.

## 4.3 Comparison with E loss

In this section, we compare the training of NQS using our proposed method with a more standard approach that utilizes the energy as a loss function (with and without SR). In all cases,

Table 1: Hyperparameters table.

| Hyperparameter | symbol | value |
|---|---|---|
| Input patch size | $d_{\text{p}}$ | 2 |
| Patch encoding size | $d_{\text{enc}}$ | 8 |
| Training batch size | $N_{\text{b}}$ | 256 |
| Initial learning rate | $\alpha_0$ | $10^{-3}$ |
| Final learning rate | $\alpha_{\text{f}}$ | $10^{-5}$ |
| Number of samples for energy statistics during training | - | $10^5$ |
| Number of samples for final energy estimation | - | $10^6$ |
| Total number of optimizer steps | - | $5 \cdot 10^5$ |
| MCMC warm-up steps | $N_{\text{warmup}}$ | $10d_{\text{lat}}^2$ |
| MCMC sample skip interval | $N_{\text{skip}}$ | 4 |

we investigate the performance on $6 \times 6$ lattice, while keeping all shared hyperparameters unchanged. The only exceptions are for SR, where we employ vanilla stochastic gradient descent instead of ADAM and increase the learning rate by a factor of 10. Due to the longer optimization time required for SR (which varies from 4 to 6 times due to the iterative solver used), we also reduce the total number of optimizer steps by a factor of 5, bringing it down to $10^5$. The energy error is calculated by comparing the achieved energy with ED results.

Figure 3(a) shows three energy minimization curves for each loss, with each curve representing different random initializations of the NQS variational parameters. It is evident that training NQS with E loss and vanilla gradient is highly unstable since the minimization curves exhibit sudden jumps in energy and converge to significantly different energy values depending on initialization. In contrast, with ITE loss there are no abrupt energy jumps and the final energy value has almost no dependence on the initial variational parameters. Moreover, employing the ITE loss consistently leads to a lower final energy for the NQS compared to the E loss approach. This instability of training with E loss likely arises because it is equivalent to ITE loss with a constantly changing target (see Sec. 2.5).

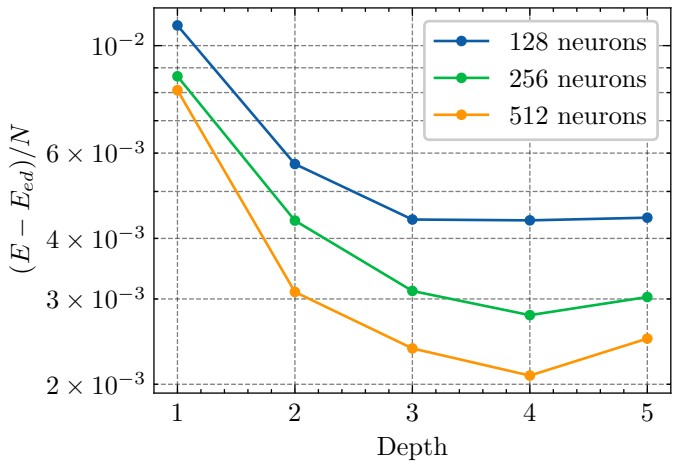

Figure 2: Energy error dependence on neural network width (number of neurons per hidden layer) and depth (number of hidden layers) for NQS trained with ITE loss. The lattice consists of $6 \times 6$ sites, and $J_2/J_1 = 0.5$. The variable $N$ represents the number of lattice sites.

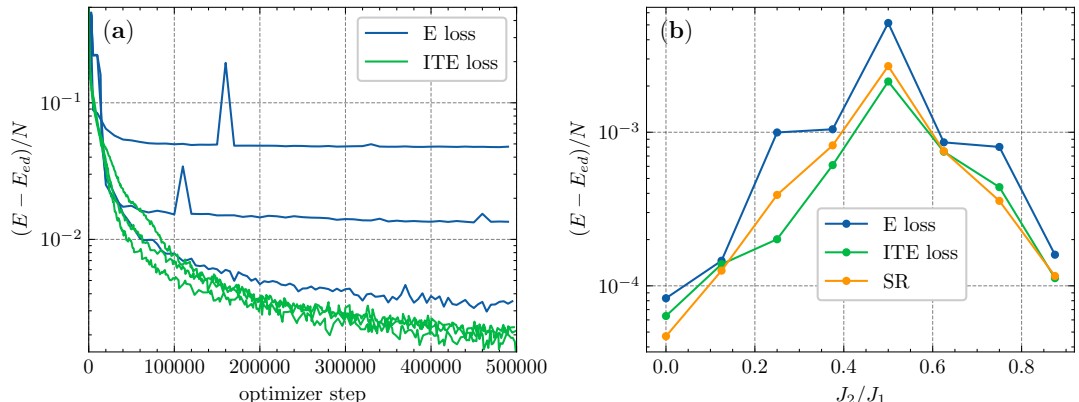

Figure 3: **(a)** Energy minimization curves for NQS trained with E loss (blue) and ITE loss (green). Each loss function has three curves, representing different random initializations of NQS variational parameters. $J_2/J_1 = 0.5$ for this analysis. **(b)** Energy error dependence on $J_2/J_1$ for NQS trained with E loss (blue), ITE loss (green), and E loss with SR (orange). Both figures are based on a lattice size of $6 \times 6$. The variable $N$ represents the number of lattice sites.

Figure 3(b) depicts the energy error's dependence on $J_2/J_1$. Given that the final energy with E loss exhibits significant variation from run to run, we conduct 5 runs for each $J_2/J_1$ value and only plot the best result. The figure clearly illustrates that our proposed method achieves superior accuracy compared to E loss with vanilla gradient not only in the vicinity of the maximum frustration point at $J_2/J_1 = 0.5$ but also over a wider parameter range. Our method also demonstrates competitiveness with SR, achieving slightly lower energy errors at some points and slightly higher errors at others. With all losses, a distinct trend is visible: the energy error reaches its maximum near $J_2/J_1 = 0.5$ and subsequently decreases as it moves further from this point. However, the trend is noisier in the case of E-loss due to the instability of convergence.

In terms of computational time, the proposed scheme generally takes approximately 30% - 50% longer in practice than training with E loss and vanilla gradient. The primary reason for this increase is the additional energy estimation performed after each optimizer step to determine when to update the target wave function (see Sec. 2.6). The calculation of the loss function requires a similar amount of computation in both cases. This is because energy estimation in E loss and target computation in ITE loss involves computing the same terms, namely $\sum_{\mathbf{s}'} H_{\mathbf{s}\mathbf{s}'}\psi(\mathbf{s}')$, which dominate the computational cost. Compared to SR, our proposed method took, on average, about 5 times less computational time per iteration. However, SR iterations seem to be more effective in reducing energy, as the final energy accuracy achieved in similar computation times (despite SR having 5 times fewer iterations) is comparable.

## 4.4 Benchmarking

In this section, we benchmark NQS trained with our proposed method against the well-established DMRG method. We do this by investigating the relationship between the predicted ground state energy and lattice size, specifically when exceeding the practical modeling limits of ED. Additionally, we compare some of our results with those achieved in other studies.

Figure 4 shows the predicted ground state energy dependence on lattice border length, while keeping $J_2/J_1 = 0.5$. Since the performance of DMRG strongly depends on the number of bond dimensions, $\chi$, we present the data for two values: 128 and 1024. Regarding the NQS, we observed that extending the training duration significantly can lead to a slight improvement

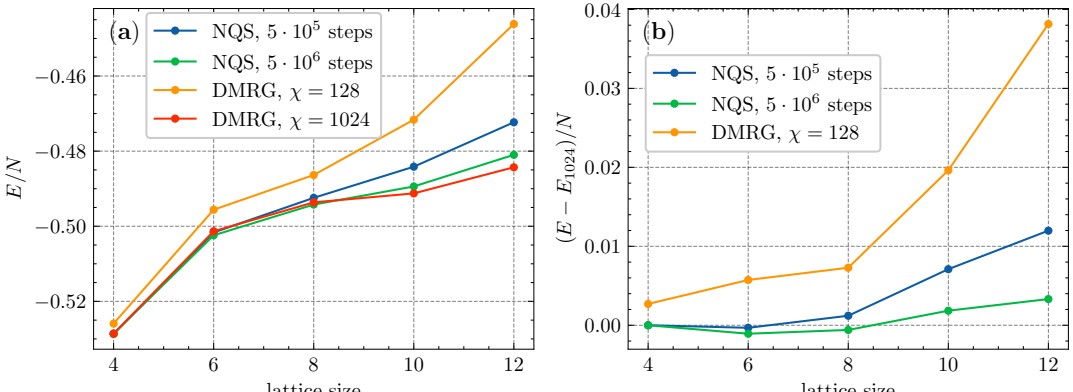

Figure 4: **(a)** Dependence of the variational ground state energy on lattice size for NQS (ITE loss) and DMRG. The blue and green lines represent NQS trained for $5 \cdot 10^5$ and $5 \cdot 10^6$ optimizer steps, respectively. The orange and red lines correspond to DMRG with 128 and 1024 bond dimensions, respectively. In this context, $J_2/J_1 = 0.5$, $N$ denotes the number of spins, and 'lattice size' refers to the border length of a square-shaped 2D lattice. **(b)** Same as (a), but with energy measured relatively to the variational energy of DMRG with 1024 bond dimensions.

in the predicted energy. As a result, we also present the data for NQS with $5 \cdot 10^6$ optimizer steps, which is an order of magnitude higher than our default value. This decrease in final energy becomes increasingly prominent with larger lattice sizes.

From the presented data, it is evident that NQS clearly outperforms DMRG with $\chi = 128$. However, when $\chi = 1024$, the situation becomes more intricate. Both NQS and DMRG achieve practically the same energy as ED for the $4 \times 4$ lattice. For the $6 \times 6$ and $8 \times 8$ lattices, NQS slightly outperforms DMRG, but only when using the extended training time. However, for larger lattice sizes beyond $8 \times 8$, DMRG starts to outperform NQS, and the difference appears to grow with increasing lattice size. In the matrix product state used in DMRG, the number of variational parameters grows proportionally with the lattice size. On the other hand, for NQS, the number of variational parameters remains almost constant as it increases only in the first hidden layer of the network, which represents a small fraction of the total number of parameters. This disparity in the scaling of variational parameters could provide an explanation for why DMRG eventually outperforms NQS with increasing lattice size.

Regarding the computational resources, it is hard to compare since we utilized CPU for DMRG and GPU for NQS. However, the actual time it took to compute the models was on a similar scale for DMRG with $\chi = 1024$ and NQS with $5 \cdot 10^6$ optimizer steps (e.g. about 20 hours for $12 \times 12$ lattice).

For $J_2/J_1 = 0.5$ and $6 \times 6$ lattice, Ref. [31] identified an energy error limit that appears to be hard to overcome for NQS. Multiple other works [58, 71, 72], including theirs, have achieved energy errors comparable to $2 \cdot 10^{-3}$, despite using different NQS architectures and optimization procedures. In Fig. 3 (b) one can see that our method also achieves a similar value at $J_2/J_1 = 0.5$. After increasing the total number of optimizer steps to $5 \cdot 10^6$, our method achieves $1.4 \cdot 10^{-3}$, which is still comparable. Here we used neural networks with a substantially larger number of variational parameters than the mentioned studies and still were unable to significantly surpass the $2 \cdot 10^{-3}$ limit. This adds more evidence that the problem is related to the variational landscape rather than the representation power of neural networks.

Several studies have explored the $J_2 - J_1$ model using NQS. In Table 2, we present some of the best energies per site achieved with a $J_2/J_1 = 0.5$ on a $10 \times 10$ lattice. All the other studies mentioned utilized translation-invariant CNNs and incorporated various symmetries

Table 2: Comparison of energy estimates with NQS for $10 \times 10$ lattice with $J_2/J_1 = 0.5$.

| Method | $E/N$ |
|:---:|:---:|
| CNN, REMD [73] | −0.4736 |
| CNN, SR [58] | −0.4952 |
| CNN, SR [74] | −0.4958 |
| CNN, minSR [75] | −0.4976 |
| MLP, ITE (ours) | −0.4894 |

of the physical system. In our study, we achieved competitive results with a straightforward and generic MLP architecture, even though we used gradient descent with Euclidean metric and did not explicitly integrate symmetries into NQS.

## 5 Conclusions

In this study, we proposed and analyzed an adaptive scheme for searching the ground state with NQS, built upon the principle of imaginary time evolution. In this method, we construct the target wave function by combining the current wave function with the discretized flow obtained from the Schrödinger equation with imaginary time. Subsequently, we train the neural network to approximate this target wave function. Through repeated iterations of this process, the state approximated by the neural network converges to the ground state. We employ a loss function based on the overlap of wave functions, utilize an adaptive, optimal imaginary time step and also fix the target wave function for multiple time steps, until a decrease in energy is observed. Differently from the commonly used SR approach, our method uses the vanilla gradient with Euclidean metric.

In our exploration of the relationship between our proposed scheme and the direct minimization of energy, we uncovered a potential explanation for the unstable convergence often observed when employing energy as a loss function. Using the energy loss is equivalent to our method but with a target that changes after every optimizer step. The instability might arise due to a constantly shifting target. In our approach, this issue is addressed by fixing the target for some number of optimizer steps, resulting in a more stable training process.

Our investigation included numerical experiments with the $J_1$–$J_2$ Heisenberg model on a 2D square lattice, providing compelling evidence that our method offers higher stability and final energy accuracy than the optimization of NQS with energy as a loss function. Moreover, it showcases competitiveness with the well-established DMRG method and NQS optimization with SR.

We emphasize that our numerical experiments were performed without leveraging the symmetries of the physical system, which significantly increases the difficulty of the problem. Yet by utilizing neural networks with a large number of parameters compared to the commonly used values in this research area, we managed to achieve comparable accuracy to the results reported in works that do exploit symmetries and use specialized neural network architectures. This highlights the primary advantage of our proposed method: the capability to utilize large neural networks and apply standard optimization methods from other areas of machine learning.

# Acknowledgments

The authors express their gratitude to Julius Ruseckas and Artūras Acus for discussions that prompted new insights. Furthermore, the authors extend their thanks to Marin Bukov for valuable correspondence regarding QuSpin, and to Giuseppe Carleo for useful comments.

**Funding information** This work was performed under the "Universities' Excellence Initiative" programme.

## A Energy moments

First, we show that the mean value corresponding to any operator, $\hat{A}$, is equal to the mean of its local value $A_{loc}(\boldsymbol{s}) := \sum_{\boldsymbol{s}'} \frac{\psi(\boldsymbol{s}')}{\psi(\boldsymbol{s})} A_{\boldsymbol{s}\boldsymbol{s}'}$:

$$
\begin{aligned}
\langle A \rangle = \langle \psi | \hat{A} | \psi \rangle &= \sum_{\boldsymbol{s}} \sum_{\boldsymbol{s}'} \psi^*(\boldsymbol{s}) \psi(\boldsymbol{s}') A_{\boldsymbol{s}\boldsymbol{s}'} \\
&= \sum_{\boldsymbol{s}} |\psi(\boldsymbol{s})|^2 \sum_{\boldsymbol{s}'} \frac{\psi(\boldsymbol{s}')}{\psi(\boldsymbol{s})} A_{\boldsymbol{s}\boldsymbol{s}'} = \left\langle \sum_{\boldsymbol{s}'} \frac{\psi(\boldsymbol{s}')}{\psi(\boldsymbol{s})} A_{\boldsymbol{s}\boldsymbol{s}'} \right\rangle_{\boldsymbol{s}} \\
&= \langle A_{loc}(\boldsymbol{s}) \rangle_{\boldsymbol{s}} .
\end{aligned}
\tag{A.1}
$$

By substituting $\hat{A} = \hat{H}$, we obtain Eq. (12).

Now we show that the mean value corresponding to the product of any two operators, $\hat{A}$ and $\hat{B}$, can also be computed with their local values:

$$
\begin{aligned}
\langle \hat{A} \hat{B} \rangle &= \sum_{\boldsymbol{s}'} \sum_{\boldsymbol{s}''} \psi^*(\boldsymbol{s}') \psi(\boldsymbol{s}'') (AB)_{\boldsymbol{s}'\boldsymbol{s}''} \\
&= \sum_{\boldsymbol{s}'} \sum_{\boldsymbol{s}''} \psi^*(\boldsymbol{s}') \psi(\boldsymbol{s}'') \sum_{\boldsymbol{s}} A_{\boldsymbol{s}'\boldsymbol{s}} \hat{B}_{\boldsymbol{s}\boldsymbol{s}''} \\
&= \sum_{\boldsymbol{s}} |\psi(\boldsymbol{s})|^2 \sum_{\boldsymbol{s}'} \frac{\psi^*(\boldsymbol{s}')}{\psi^*(\boldsymbol{s})} A_{\boldsymbol{s}'\boldsymbol{s}} \sum_{\boldsymbol{s}''} \frac{\psi(\boldsymbol{s}'')}{\psi(\boldsymbol{s})} \hat{B}_{\boldsymbol{s}\boldsymbol{s}''} \\
&= \sum_{\boldsymbol{s}} |\psi(\boldsymbol{s})|^2 A^*_{loc}(\boldsymbol{s}) B_{loc}(\boldsymbol{s}) \\
&= \langle A^*_{loc}(\boldsymbol{s}) B_{loc}(\boldsymbol{s}) \rangle .
\end{aligned}
\tag{A.2}
$$

By substituting $\hat{A} = \hat{B} = \hat{H}$, we obtain Eq. (13). By susbstituting $\hat{A} = \hat{H}$ and $\hat{B} = \hat{H}^2$ we obtain Eq. (14).

In both derivations, we made the simplifying assumption of a normalized state, $|\psi\rangle$. However, it is clear that the same steps can be performed without this assumption, resulting in identical final expressions.

## B Gradient of ITE loss

First we split the gradient into two terms:

$$
\frac{\partial L}{\partial \theta} = -\frac{\partial}{\partial \theta} \log \frac{|\langle \psi | \psi_T \rangle|^2}{\langle \psi | \psi \rangle \langle \psi_T | \psi_T \rangle} = \frac{\partial}{\partial \theta} \log \langle \psi | \psi \rangle - \frac{\partial}{\partial \theta} \log |\langle \psi | \psi_T \rangle|^2 ,
\tag{B.1}
$$

then calculate each term separately:

$$
\begin{aligned}
-\frac{\partial}{\partial\theta}\log|\langle\psi|\psi_T\rangle|^2 &= -2\Re\left\{\frac{\partial}{\partial\theta}\log\langle\psi|\psi_T\rangle\right\}\\
&= -2\Re\left\{\frac{1}{\sum_{s'}\psi^*(s')\psi_T(s')}\sum_s\psi_T(s)\frac{\partial}{\partial\theta}\psi^*(s)\right\}\\
&\quad -2\Re\left\{\frac{1}{\sum_{s'}|\psi(s')|^2\cdot\frac{\psi_T(s')}{\psi(s')}}\sum_s|\psi(s)|^2\frac{\psi_T(s)}{\psi(s)}\frac{\partial}{\partial\theta}\log\psi^*(s)\right\}\\
&\quad -2\Re\left\{\frac{1}{\left\langle\frac{\psi_T(s)}{\psi(s)}\right\rangle_s}\left\langle\frac{\psi_T(s)}{\psi(s)}\frac{\partial}{\partial\theta}\log\psi^*(s)\right\rangle_s\right\},
\end{aligned}
\tag{B.2}
$$

$$
\begin{aligned}
\frac{\partial}{\partial\theta}\log\langle\psi|\psi\rangle &= \frac{1}{\langle\psi|\psi\rangle}\frac{\partial}{\partial\theta}\sum_s\psi(s)\psi^*(s)\\
&= 2\Re\left\{\frac{1}{\langle\psi|\psi\rangle}\sum_s\psi(s)\frac{\partial}{\partial\theta}\psi^*(s)\right\}\\
&= 2\Re\left\{\frac{1}{\langle\psi|\psi\rangle}\sum_s|\psi(s)|^2\frac{\partial}{\partial\theta}\log\psi^*(s)\right\}\\
&= 2\Re\left\{\left\langle\frac{\partial}{\partial\theta}\log\psi^*(s)\right\rangle_s\right\}.
\end{aligned}
\tag{B.3}
$$

Finally, putting these results back into Eq. (B.1), we obtain Eq. (9).

## C  Relation between gradients of ITE loss and E loss

The ratio between the target wave function [Eq. (7)] and the current wave function is related to the local value of the Hamiltonian:

$$
\begin{aligned}
\frac{\psi_T(s)}{\psi(s)} &= \frac{\psi(s)-\Delta\tau\sum_{s'}\hat{H}_{ss'}\psi(s')}{\psi(s)}\\
&= 1-\Delta\tau\sum_{s'}\frac{\psi(s')}{\psi(s)}\hat{H}_{ss'}\\
&= 1-\Delta\tau H_{loc}(s).
\end{aligned}
\tag{C.1}
$$

By substituting this ratio into Eq. (9) we obtain Eq. (20):

$$
\begin{aligned}
\frac{\partial L}{\partial\theta} &= 2\Re\left\{\left\langle\frac{\partial}{\partial\theta}\log\psi^*(s)\right\rangle_s - \frac{1}{\left\langle\frac{\psi_T(s)}{\psi(s)}\right\rangle_s}\left\langle\frac{\psi_T(s)}{\psi(s)}\frac{\partial}{\partial\theta}\log\psi^*(s)\right\rangle_s\right\}\\
&= 2\Re\left\{\left\langle\frac{\partial}{\partial\theta}\log\psi^*(s)\right\rangle_s - \frac{\left\langle(1-\Delta\tau H_{loc}(s))\frac{\partial}{\partial\theta}\log\psi^*(s)\right\rangle_s}{\langle 1-\Delta\tau H_{loc}(s)\rangle_s}\right\}\\
&= 2\Re\left\{\frac{\Delta\tau\left\langle H_{loc}(s)\frac{\partial}{\partial\theta}\log\psi^*(s)\right\rangle_s - \Delta\tau\langle E\rangle\left\langle\frac{\partial}{\partial\theta}\log\psi^*(s)\right\rangle_s}{1-\Delta\tau\langle E\rangle}\right\}\\
&= \frac{\Delta\tau}{1-\Delta\tau\langle E\rangle}2\Re\left\{\left\langle H_{loc}(s)\frac{\partial}{\partial\theta}\log\psi^*(s)\right\rangle_s - \langle E\rangle\left\langle\frac{\partial}{\partial\theta}\log\psi^*(s)\right\rangle_s\right\}\\
&= \frac{\Delta\tau}{1-\Delta\tau\langle E\rangle}\frac{\partial L_E}{\partial\theta}.
\end{aligned}
\tag{C.2}
$$

Here in the first line we used the result from Sec. B.

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
