# Peer review of "Scalable Imaginary Time Evolution with Neural Network Quantum States"

_SciPost Physics, doi:SciPost Phys. 15, 229 (2023)_

## Round 2 · Referee Report · Anonymous (Referee 1) · 2023-10-2

Strengths

The paper contains a good analysis of a supervised approach to find ground states, that does not require computing the quantum geometric tensor. A connection between the gradient of the overlap and the standard used gradient is presented, this is also of general interest.

Weaknesses

  1. The presentation conveys the idea that the main technique presented in the manuscript (supervised learning of one step of imaginary evolution) is new. However, this is not the case (see Report) and the main novelty is instead the adaptive step and fixing the target state of a few steps. The manuscript should reflect this.

Report

I would recommend publication of this work in SciPost, since the work contains a nice analysis of the supervised (approximate) imaginary-time evolution and introduces an interesting adaptive method.

However, I believe the paper should be rewritten in key places in order to convey more clearly that the key novelty here is the adaptive scheme and not the supervised approach. To be clear, this fact, by itself, does not diminish the importance of the work, but must be addressed in the rewriting phase.

Requested changes

  1. As mentioned in the report, the authors should clarify that the supervised learning of the imaginary time evolution (or of the power-method /Euler approximation ) was pre-existing, and that the main novelty here is the adaptive step/ keeping the target fixed for a few steps. The authors do briefly acknowledge the existence of previous works exploring the same idea "We observe that several studies have relied on similar techniques to the one presented in this section within the context of unitary dynamics [28,47–49], as well as in non-unitary dynamics or power iteration [50,51]." but this mention is lost both in the abstract and in the introduction. Also, it is not mentioned that the novelty is the adaptive step.

In the introduction they also write "In the present paper, we propose a ground-state determination ... " Also, they mention the method is inspired by Ref. 28, however the more directly related methods are arguably 50 and 51, as well as H.Atanasova, et al. Nature Communications 14 (2023) . doi:10.1038/s41467-023-39244-4.

  1. The authors compare their approach to standard gradient-descent based on the gradient of the energy and see a nice improvement. Do the authors use standard SGD or a better optimizer such as Adam? It would be important to include at least something beyond bare SGD

  2. Related to the previous point, and in order to show that their method converges to an energy at least similar to what obtained by using the more commonly adopted Stochastic Reconfiguration (SR), it would be important that the authors add a curve based on SR , in Figure 3(b). SR is implemented in standard open source packages (NetKet for example) and does not require major implementation steps on their side.

  3. When discussing the comparison to DMRG, it is important to mention that state-of-the-art optimization techniques and architectures can significantly outperform DMRG, also beyond 8x8. I would encourage the authors for example to consider these works https://arxiv.org/pdf/2302.01941.pdf , https://arxiv.org/abs/2104.05085, https://arxiv.org/abs/2301.06788 , Ref 66, also considering that Ref 56 and 71 are rather "old" and do not reflect the state of the art anymore.

  • validity: good
  • significance: ok
  • originality: good
  • clarity: ok
  • formatting: good
  • grammar: excellent

Author:  Eimantas Ledinauskas  on 2023-10-17  [id 4042]

(in reply to Report 1 on 2023-10-02)

Thank you for your comments and the suggested papers.

We have made revisions to the third paragraph of the introduction and have also adjusted the abstract and conclusions to clearly acknowledge that the concept of supervised learning of imaginary time evolution was pre-existing. In addition, we made minor modifications throughout the text to ensure consistency.

Regarding the comment about SGD and ADAM, we already employ ADAM, as indicated in Sec. 4.2. To enhance its visibility, we have added additional references to ADAM in other sections.

In response to your suggestion, we have incorporated the SR curve into Fig. 3(b) and provided commentary on it in Sec. 4.3.

Furthermore, as recommended, we have included more results for the 10x10 lattice, which better represent the current state of the art in Sec. 4.4. These energy results are now summarized in Table 2.

---

## Round 3 · Referee Report · Anonymous (Referee 3) · 2023-11-14

Report

In this manuscript the authors introduce a novel way to realize imaginary time evolution for a recently emerging numerical method called neural quantum states. Such imaginary time evolution is crucial when aiming to find the ground state of quantum matter, one key goal in computational physics. In this context the current manuscript is very timely. Further, the manuscript is also very well written and easily accessible also to non-experts. I particularly appreciate the care and extensive discussion on the numerical aspects such as the inclusion of all the hyperparameters for their simulations.

Overall, I therefore recommend publication in Scipost.

I just have a few suggestions and questions, which might be worthwhile to address:

1. When I understood correctly, Eq. (7) describes how the authors perform the time evolution before they do the optimization based on the loss function in Eq. (8). Is the current wave function obtained through iterating Eq. (7) for a few time steps? If yes, is this something which can be controlled for large system sizes? I'm asking because the wave function amplitudes psi(s) should typically behave exponentially on system size, which might make Eq. (7) impossible to handle because of over- or underflow errors due finite accuracy. Is this something the authors would agree to or maybe they have observed this even already?

2. The computational effort of the present imaginary time approach is proportional to N^2 with N the number of degrees of freedom because higher-order moments of the energy need to be computed. Do the authors also know to which extent this also means that the number of Monte Carlo samples has to be increased? I guess that higher-order moments require more samples for the same accuracy.

  • validity: top
  • significance: high
  • originality: high
  • clarity: top
  • formatting: perfect
  • grammar: perfect

Author:  Eimantas Ledinauskas  on 2023-11-17  [id 4125]

(in reply to Report 2 on 2023-11-14)
Category:
answer to question

Thank you for your comments and suggestions. Below are our replies:

  1. The target wave function is calculated using only a single time step. Performing a second time step is not practical for large systems, as it would necessitate an additional application of the Hamiltonian to the resulting target wave function. This would change the scaling from linear to quadratic with respect to system size (as similarly described in Eq. 16-17). Additionally, as you have pointed out, the increasing number of summation terms could pose problems with numerical accuracy. However, this issue is not relevant in our case, because we only require a single time step. We added this clarification in text after Eq. 7.

  2. This is an interesting point. We have not investigated in detail whether higher-order moments require more samples to achieve the same level of accuracy. However, the accuracy of moment estimations can always be monitored by calculating the standard error of the sample. Additionally, this error diminishes as the variational wave function approaches the ground state (where, with the exact ground state, every sample point would have the same local energy value). Consequently, in practice, we have not encountered any problems related to the accuracy of higher-order moments. We note that the computational complexity for all three energy moments utilized in this work actually scales linearly with the system's size. This linear scaling is due to the fact that the average of E^2 can be calculated from a sample of first-order H_loc values (as shown in Eq. 13 and derived in Eq. 23). For the average of E^3, a sample of second-order H_loc values (Eq. 14) would be required, leading to quadratic scaling. However, as described at the end of Sec. 2.4, we instead use an approximate estimate (Eq. 18) which also scales linearly. While this estimate might be biased, our numerical experiments suggest that it is likely sufficient. We added this clarification in text after Eq. 18.

---

## Round 3 · Referee Report · Anonymous (Referee 2) · 2023-11-14

Report

I thank the authors for the work done in revising their manuscript and improving the presentation. I believe the paper is now ready to be accepted for publication.

---

## Round 3 · Author Response

We thank Referee 1 for comments and the suggested papers.

We have made revisions to the third paragraph of the introduction and have also adjusted the abstract and conclusions to clearly acknowledge that the concept of supervised learning of imaginary time evolution was pre-existing. In addition, we made minor modifications throughout the text to ensure consistency.

Regarding the comment about SGD and ADAM, we already employ ADAM, as indicated in Sec. 4.2. To enhance its visibility, we have added additional references to ADAM in other sections.

In response to Referee 1 suggestion, we have incorporated the SR curve into Fig. 3(b) and provided commentary on it in Sec. 4.3.

Furthermore, as recommended, we have included more results for the 10x10 lattice, which better represent the current state of the art in Sec. 4.4. These energy results are now summarized in Table 2.

---

## Round 4 · Author Response

In response to Referee 2's comments, we added minor clarifications in text right after Eq. 7. and Eq. 18.

---

## Editorial Decision

published